# Patello-Femoral Pain Syndrome: Magnetic Resonance Imaging versus Ultrasound

**DOI:** 10.3390/diagnostics13081496

**Published:** 2023-04-21

**Authors:** Patrizia Pacini, Milvia Martino, Luca Giuliani, Gabriele Santilli, Francesco Agostini, Giovanni Del Gaudio, Andrea Bernetti, Massimiliano Mangone, Marco Paoloni, Martina Toscano, Corrado De Vito, Carlo Ottonello, Valter Santilli, Vito Cantisani

**Affiliations:** 1Policlinico Umberto I Hospital, Department of Radiological and Oncological Sciences and Pathological Anatomy, Sapienza University, Policlinico Avenue 155, 00161 Rome, Italy; patry.shepsut91@gmail.com (P.P.); aivlim@hotmail.it (M.M.); g.d.gaudio@gmail.com (G.D.G.); toscmartina@gmail.com (M.T.); corrado.devito@uniroma1.it (C.D.V.); vito.cantisani@uniroma1.it (V.C.); 2San Salvatore Hospital, Department of Biotechnological and Applied Clinical Sciences, University of L’Aquila, Vetoio Stree, 67100 L’Aquila, Italy; 3Policlinico Umberto I Hospital, Department of Anatomical, Histological and Legal Medical Sciences and of Locomotor System, Sapienza University, Aldo Moro Square 3, 00185 Rome, Italy; gabriele.santilli@uniroma1.it (G.S.); francesco.agostini@uniroma1.it (F.A.); andrea.bernetti@uniroma1.it (A.B.); massimiliano.mangone@uniroma1.it (M.M.); marco.paoloni@uniroma1.it (M.P.); valter.santilli@uniroma1.it (V.S.); 4Fisiocard Medical Centre, Via Francesco Tovaglieri 17, 00155 Rome, Italy; carlo.ottonello@gmail.com

**Keywords:** PFPS, MRI, US

## Abstract

Background: Magnetic Resonance Imaging (MRI) and Ultrasound (US) in combination with clinical data could contribute to the diagnosis, staging and follow-up of Patello-Femoral Syndrome (PFS), which often overlaps with other pathologies of the knee. Purpose of the Study: To evaluate the diagnostic role of MRI and US findings associated with PFS and define the range values of instrumental measurements obtained in pathological cases and healthy controls, the performance of the two methods in comparison, and the correlation with clinical data. Materials and Methods: 100 subjects were examined: 60 patients with a high suspicion of PFS at the clinical evaluation and 40 healthy controls. All measurements obtained by MRI and US examination were correlated with clinical data. A descriptive analysis of all measurements was stratified for pathological cases and healthy controls. A Student’s *t*-test for continuous variables was used to compare patients to controls and US to MRI. Logistic regression analysis was applied to test the correlation between MRI and US measurements with clinical data. Results: Statistical descriptive analysis determined the MRI and US range values of medial patello-femoral distance and the thickness of retinacles and cartilages in pathological cases and healthy controls. In pathological cases, the retinacle results of both increased; the medial appeared to be slightly more increased than the lateral. Furthermore, in some cases, the thickness of the cartilage decreased in both techniques; the medial cartilage was more thinned than the lateral. According to logistic regression analyses, the best diagnostic parameter was the medial patello-femoral distance due to the overlapping results of the US and MRI. Furthermore, all clinical data obtained by different tests showed a good correlation with patello-femoral distance. In particular, the correlation between medial patello-femoral distance and the VAS score is direct and equal to 97–99%, which is statistically significant (*p* < 0.001), and the correlation with the KOOS score is inverse and equal to 96–98%, which is statistically significant. Conclusions: MRI and Ultrasound examination in combination with clinical data demonstrated high-value results in the diagnosis of PFS.

## 1. Introduction

Patello-Femoral Pain Syndrome (PFPS, also referred to as chondromalacia patella, anterior knee pain, and runner’s knee) is a frequent pathological condition of the knee typical of young adults, defined by diffuse peri- or retro-patellar pain in the anterior and/or medial aspect of the knee that is worsened by activities performed with the flexed knee (e.g., squats, climbing or descending stairs) [1,2]. Although in the past it was often undiagnosed or confused with other diseases, PFPS is currently a frequently recognized condition.

It is a chronic disease caused by overuse and misuse, rather than acute trauma, and it is broadly classified into two categories: patellar malalignment and patellar maltracking [3].

It has a prevalence of 7–28% and an incidence of 9.2% [1]. This syndrome occurs more frequently in the female sex, with a ratio of 2:1 compared to the male sex, and in the sporting population (25–40%) [2,3,4]. It has been postulated that PFPS has a static and dynamic multifactorial etiology [5,6,7,8,9,10], including a number of factors that cause an excessive stress at the patello-femoral joint, such as errors or overloads during training [5], the morphological alteration of the femoral trochlea and patella, the strength deficit of the vastus medialis muscle, and the altered timing of the activation of the vastus medialis oblique muscle. However, it seems that a key factor in the genesis of PFPS may be dynamic knee valgus, which would cause patellar abnormal lateral movement. Symptoms of PFPS may persist for years after onset if not adequately treated, and this could limit the patient’s physical activity [11,12]. 

Indeed, the diagnosis of PFPS is predominantly clinical and is based mainly on the history of the patient and a thorough physical examination [11]. Many studies state that the clinical suspicion of PFPS is confirmed by various imaging techniques such as X-ray, which is commonly the first imaging examination for assessing morphological changes in the evaluated bone segments; CT, also used for the evaluation of patellar and trochlear morphology; and MRI, used in particular for its contrast resolution and multiplanar capability for the knee joint and the study of soft tissue [13,14].

Although the value of ultrasonography examination of the knee is well known, this imaging technique is not sufficiently considered for the diagnosis of specific pathologies of this anatomic district; in particular, one paper reports that US is very useful in the diagnosis of anterior knee pain [15]. Ultrasound (US) allows for the evaluation of the peripatellar soft tissues and the trochlear cavity cartilage, as well as its relationship with the patellar tip. Quantitative and qualitative US criteria have been approved as an important tool in any diagnostic testing [16,17], but the reliability of their inter-operator measurements has never been studied. Compared to most radiological techniques, in which images are acquired using predefined protocols, US image acquisition is more complex and operator-dependent [18].

Therefore, the aim of the present study is to demonstrate how MRI and US investigations, mainly in combination, could have the same diagnostic value in the static phase for the diagnosis of patello-femoral instability, if the examination is performed by an experienced operator taking precise measurements of certain knee structures to complete the clinical information for diagnosis purposes. However, given the few studies performed in the literature, this needs more confirmation in terms of the reliability of the results. 

## 2. Materials and Methods

During the period between March 2021 and July 2022, 72 patients were examined for chronic (>3 months) anterior knee pain. A total of 100 subjects were enrolled in the study: 60 patients with a high suspicion or confirmed diagnosis of PFS at the clinical exam and 40 healthy controls. MRI and US were performed to evaluate medial patello-femoral distance and the thickness of retinacles and cartilages (Figure 1, Figure 2, Figure 3 and Figure 4). The alterations of the following anatomical parameters/structures are evaluable in both exams applied and are taken into account in PFS for the choice of conservative or surgical therapy: patellar malalignment, retinacle alterations and cartilages damage [3]. A quantitative analysis was conducted to compare all instrumental measurements with score data collected by the use of clinical questionnaires.

### 2.1. Physiatric Tests

In the clinical physiatric evaluation, the patients described the anterior knee or peri-/retro-patellar pain that occurs during physical activities or positions with the knee flexed, such as a squat, walking, running, jumping and sitting, perhaps also reporting sensations of crepitus [1]. 

The physical examination included an evaluation of the patient in the standing position and a dynamic examination with an evaluation in supine, lateral, and prone positions [3].

To estimate the severity of pain, a visual analogue scale (VAS) was used.

Patients who had positive results after the following anterior knee pain provocation tests for the suspicion of PFPS and with a VAS score > 3 were included in the study:-Movie Theater Sign: consists of anterior pain of the knee when standing after a prolonged period of sitting;-A simple squat, with pain in 80% of patients [1,2,3,4,5,6,7,8,9,10,11];-Palpation of the margins and facet joints of the patellae (71–75% of subjects with this sign have PFPS);-Patellar Apprehension Sign;-Grind test (Clarke’s sign or Zohler’s sign): Patient is in a supine or long sitting position with the involved knee extended. The physician applies pressure with the hand superior to the patella while the patient gradually contracts the quadriceps muscle. The test is positive with the presence of pain in the patello-femoral joint [1];-Stair test: consists of pain triggered by walking up and down stairs.

All subjects underwent a clinical quantitative evaluation by several questionnaires: VAS score, KOOS score, function daily living, function sports and recreational activity, symptoms stiffness, and quality of life.

The exclusion criteria were: (i) patients who underwent knee surgery in the last 10 years, (ii) subjects with neurological or rheumatologic conditions, (iii) patients who underwent MRI examination and not US examination, (iv) diabetes, (v) neoplasia, and (vi) taking analgesic medications.

### 2.2. Ultrasound

A B-mode Ultrasound examination was performed using a Canon Aplio I-800 scanner (from Medical Systems Corporation, Otawara, Japan) with a high-frequency (4–15 MHz) linear probe. The patients were in supine position with their knee flexed to 45°. An expert radiologist measured the medial and lateral sides of the trochlear cartilage on the axial plane, the retinacle thickness on the long axis, and the femoro-patellar distance from the medial patellar margin to the medial femoral condyle. All measurements were taken on the best plane of visualization of the structures involved.

US examination was performed by a radiologist with over 5 years of experience who was blinded to MRI and clinical data.

### 2.3. MRI

MRI was performed with a 0.25 T Esaote S-scan (from Genova, Italy), after 1 week by ultrasonography examination, with the acquisition of T1, T2-weighted fat-suppressed, and not-fat-suppressed sequences from the static condition of the knee.

The MRI examination was performed by a radiologist with over 5 years of experience that was blind to clinical data and to US results.

The subjects were positioned with their knee flexed at 45° and with a dedicated coil.

The T1–T2 weighted images were acquired on sagittal, axial, and coronal planes. A sagittal STIR sequence was also acquired. All images extended proximally to the level of the superior pole of the patella and distally at the level of the tibial tubercle. The measurements were taken mainly in the axial plane. For the patellar femoral distance, we considered an intermediate plane with the best visualization of both condyles, and the same was true for the thickness of cartilages and retinacles.

### 2.4. Statistical Analysis

All of the subjects included in the present study underwent US and MRI exams in order to perform descriptive analysis for: the thickness of retinacles (mean and median values), trochlear cartilage (mean and median values), and femoral-rotula medial distance (mean and median values).

A Student’s *t*-test for continuous variables was used to compare differences in characteristics between patients and control subjects. Otherwise, a Mann–Whitney U test was applied. In addition, logistic regression analyses were applied to test potential differences when comparing US and MRI measurements with clinical data. *p* < 0.05 was considered statistically significant.

## 3. Results

A total of 60 patients with a high suspicion of PFP at clinical evaluation and 40 control subjects were included in the study. The age range was from 20 to 60 years old (mean age: 51 years old).

Statistical descriptive analysis determined the MRI and US range values of medial patello-femoral distance and the thickness of retinacles and cartilages in pathological cases and healthy controls (Table 1).

A Student’s *t*-test was applied for each parameter considered to evaluate the media differences values, comparing cases and controls and the different performances of MRI and US.

The thickness of retinacles appeared to be more increased (mainly the medial retinaculum) in cases than controls and in both techniques performed (Table 2A). In addition, MRI and US showed in pathological cases a statistically significant difference, with retinacle thickness values recorded after MRI being slightly lower than US (Table 2B). In pathological cases, the thickness of cartilage appeared to be decreased in comparison to healthy controls, in particular on medial side. MRI showed a statistically significant difference for cases in comparison to US, with the media values obtained being slightly higher than US (Table 3). Medial patello-femoral distance in cases and controls showed a statistically significant difference, with increased values in pathological subjects (Table 4A). The media values of MRI and US showed no statistically significant differences in cases with overlapped results (Table 4B).

Descriptive statistical analysis was also performed for clinical data (Table 5). A Student’s *t*-test showed a statistically significant difference in the media values of cases compared to controls (Table 6). The most interesting data among the parameters analyzed were represented by the medial patello-femoral distance, with overlapping results between US and MRI in patients with PFS. A logistic regression analysis showed a high correlation between medial patello-femoral distance and clinical data; in particular, the VAS score results were direct and equal to 97–99%, and the KOOS scores were inverse and equal to 96–98%, and both were statistically significant (*p* < 0.001) (Table 7).

## 4. Discussion

PFPS is defined as anterior knee pain involving the patella, retinaculum, and adjacent soft tissues, after excluding the intra-articular pathology of the knee.

The most commonly believed etiologies of pain in PFPS are chondromalacia and retinacular pain [19,20,21,22,23,24,25]. The diagnosis of PFPS remains difficult [26]. In current practice, the diagnosis is primarily based on the clinical presentation of the patient. In this study a relatively large number of patients with a high suspicion of PFPS were compared to a relatively homogeneous case control population. The aim of this study was based on the evaluation of the efficacy of MRI and US compared with clinical data to identify structural changes in retinacle thickness, cartilage, and medial patello-femoral distance, which are taken into account in PFS regarding the choice of conservative or surgical therapy [3]. In pathological cases, the statistical stratification in quartiles provided the classification into severe, medium, and mild entity grades. A Student’s *t*-test showed the increase in retinacle thickness, mainly the medial retinaculum, in pathological cases compared to healthy controls and in both techniques. In addition, US showed a statistical difference compared to MRI, demonstrating slightly higher values in pathological cases. A recent study of Tyler M. Coupal (2018) [27] showed an increased retinacle thickness when assessed by the MRI method due to the strain effect of patellar lateralization.

A study conducted by Schoots EJM et al. (2013) [28] showed the presence of structural changes in the lateral retinaculum using US examination in patients with PFPS. The results of these measurements indicate a trend towards a larger thickness of the lateral retinaculum and showed neovascularization when measured by US and color-Doppler examination in patients with PFPS. The increase in thickness of the medial retinaculum maybe correlates with a chronic condition representative of the population examined in relation to fibrotic processes and a long-lasting tension state. Our study confirmed this data using MRI and US. In addition, in our study, the thickness of cartilage appears to be decreased using both techniques performed and in comparison to healthy controls. In pathological cases, the medial cartilage was more thinned than the other side in both exams performed, and compared to the healthy controls. A Student’s *t*-test showed a statistically significant difference in the performance of the two methods, with slightly higher MRI values compared to US findings. MRI has proven to be a valid, non-invasive method for the evaluation of patellar cartilage [27].

In pathological cases, the medial patello-femoral distance increased proportionally to the degree of severity and more than the healthy controls, as shown in both exams. The Student’s *t*-test showed a no statistically significant difference in the performance of the two methods with overlapped results. The abnormal position of the patella with respect to the femoral trochlear groove in the PFPS was demonstrated in several studies based on MRI results [27]. Furthermore, Lok Yin Ada Kwan in a recent study (2022) [29] showed the performance of US in the measurement of the patellar position relative to the femoral condyle, and the reliability of the results suggest that it can be introduced in current practice to assess lateral patellar displacement.

The patello-femoral distance was considered the parameter with more diagnostic value due to the overlapping of the results in both US and MRI. A logistic regression analyses was applied, and all clinical data obtained by different tests (KOOS, symptoms stiffness, function daily living, function sports and recreational activities, quality of life, VAS score) showed a good correlation with patella-femoral distance. In particular, the correlation between the medial condyle distance measured with MRI-US and the VAS scale results was direct and equal to 99% (0.9873) and was statistically significant (pV < 0.001). The correlation between the medial condyle distance measured by MRI-US and the KOOS score was inverse and equal to 98% (-0.9814) and was statistically significant.

Radiography, computerized tomographic (CT) scanning, and Magnetic Resonance Imaging (MRI) have a significant role in PFPS because they can help to rule out other related differential diagnoses during clinical evaluation, such as lateral meniscus tear, extensor tendon tear, anterior tenosynovial giant cell tumor, and plica syndrome [30,31]; however, no gold standard has been established [32,33,34,35]

Currently, there are more studies confirming the more-recognized role of MRI in comparison to US in the diagnosis of PFS. To date, despite the relevant use of US for the diagnosis of musculoskeletal disorders and the growing prevalence of PFPS, only a few papers have been published on the use of US for its identification and classification. The introduction of US represents an added value to potentiate the diagnosis of this misconstrued syndrome, mainly if in combination with another validated exam and clinical data. US has been postulated to be very effective in this syndrome, considering its low cost; widespread availability; and accuracy in evaluating the knee, especially certain peripatellar soft tissues as retinacles, the hyaline cartilage, and the relationship of this structure with the patella. A systematic review of 2015 (Fischoff C.) found that only one diagnostic study published by Lapègue et al. [18] showed a relatively simple protocol used in current practice for MRI evaluation; it is performed by US based on three diagnostic criteria (cartilaginous trochlear angle, PTTG distance, and the presence or absence of dysplasia). Moreover, quantitative and qualitative US criteria have been validated [17,18]. However, further research on the reliability of US tests for these criteria is needed. To date, despite the diagnosis of PFPS being based on the clinical presentation of the patient in some studies, as the reviews of Nunes et al. [36] and Cook et al. [37] suggest, there is no clinical test with diagnostic consistency. In a recent study of Jason A. Wallis (2021) [38], a systematic review was conducted to evaluate clinical practice guidelines for the physical therapist management of patello-femoral pain.

In this study, there are also some limitations that should be mentioned:-The sample volume of patients is small;-US examination fails to fully evaluate the patello-femoral joint because of the bony interface of the patella itself;-Currently, the US exam can be performed in a static phase;-While using both the imaging methods we could not discriminate whether the PFPS is primary or secondary to other pathologies.

This preliminary study shows that US may achieve results similar to MRI, exhibiting high agreement with clinical evaluation; therefore, we have just commenced other studies that additionally include a dynamic MRI exam and US-elastography, which will evaluate the course of the disease after undergoing for therapy 3–6 months and check this procedure’s ability to predict outcomes.

## 5. Conclusions

MRI and Ultrasound examination in combination with clinical data demonstrated high-value results in the diagnosis of PFS. Medial patello-femoral distance increased proportionally to the severity degree compared to the healthy controls, as shown in both exams. However, these preliminary results should be confirmed in view of an accurate diagnosis for a timely therapeutic approach and the follow-up of these patients.

## Figures and Tables

**Figure 1 diagnostics-13-01496-f001:**
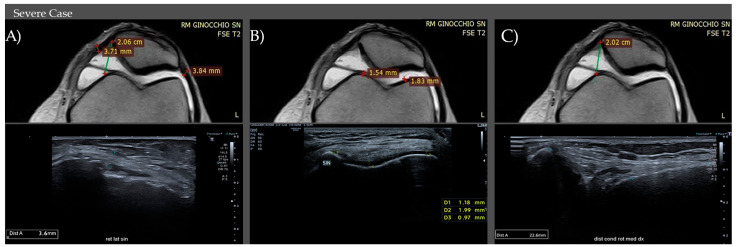
MRI and US values of retinacles (**A**) and cartilage (**B**) thickness and medial patello-femoral distance (**C**). Images of severe pathological cases.

**Figure 2 diagnostics-13-01496-f002:**
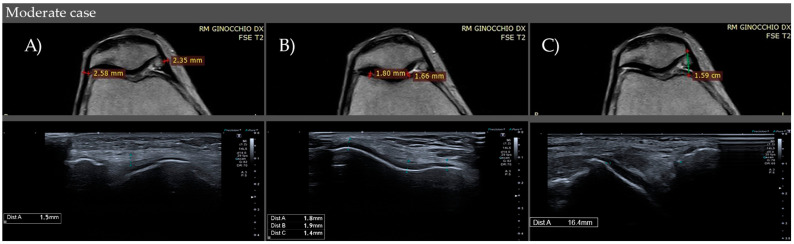
MRI and US values of retinacles (**A**) and cartilage (**B**) thickness and medial patello-femoral distance (**C**). Images of moderate pathological cases.

**Figure 3 diagnostics-13-01496-f003:**
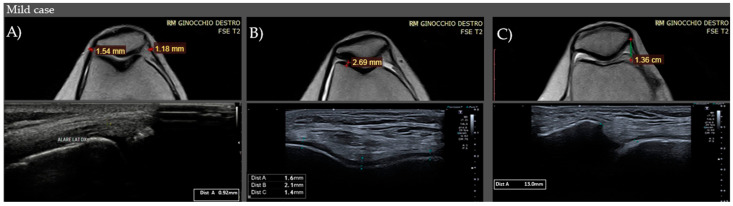
MRI and US values of retinacles (**A**) and cartilage (**B**) thickness and medial patello-femoral distance (**C**). Images of mild pathological cases.

**Figure 4 diagnostics-13-01496-f004:**
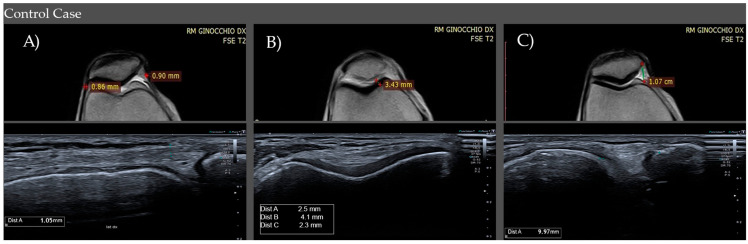
MRI and US values of retinacles (**A**) and cartilage (**B**) thickness and medial patello-femoral distance (**C**). Images of healthy controls.

**Table 1 diagnostics-13-01496-t001:** Descriptive analysis of all instrumental measurements. Median and Interquartile Range values of the parameters analyzed (thickness of retinacles and trochlear cartilages, and medial patello-femoral distance) in US and MRI in healthy controls and pathological cases, with interquartile stratification in pathological cases to differentiate the severity degree.

THICKNESS RENITACLES		MRI M (IQR)	US M (IQR)
CONTROLS	Medial Retinaculum	0.9–1.0 mm (IQR) (M1)	0.9–1.1 mm (IQR) (M1)
	Lateral Retinaculum	0.7–1.0 mm (IQR) (M0.9)	0.9–1 mm (IQR) (M1)
CASES	Medial Retinaculum	1.3–2.6 (IQR) (M2.1) Mild: <1.3 mm Moderate: 1.3–2.6 mm Severe: >2.6 mm	1.5–2.8 (IQR) (M2.3) Mild: <1.5 mm Moderate: 1.5–2.8 mm Severe: >2.8 mm
	Lateral Retinaculum	1.2–2.5 (IQR) (M0.9) Mild: <1.2 mm Moderate: 1.2–2.5 mm Severe: >2.5 mm	1.3–2.5 (IQR) (M2.2) Mild: <1.3 mm Moderate: 1.3–2.5 mm Severe: >2.5 mm
**THICKNESS CARTILAGES**		**MRI** **M (IQR)**	**US** **M (IQR)**
CONTROLS	Medial Thickness	2.8–4.7 mm (IQR) (M4)	2.9–4.5 mm (IQR) (M3.8)
	Lateral Thickness	3.3–5.1 mm (IQR) (M4.45)	3.0–4.8 mm (IQR) (M4.2)
CASES	Medial Thickness	1.5–2.3 (IQR) (M1.8) Mild: >2.3 mm Moderate: 1.5–2.3 mm Severe: <1.5 mm	1.2–2.1 (IQR) (M1.55) Mild: >2.1 mm Moderate: 1.2–2.1 mm Severe: <1.2 mm
	Lateral Thickness	1.8–2.65 (IQR) (M2.3) Mild: >2.6 mm Moderate: 1.8–2.6 mm Severe: <1.8 mm	1.65–2.5 (IQR) (M2) Mild: >2.5 mm Moderate: 1.6–2.5 mm Severe: <1.6 mm
**PATELLO-FEMORAL DISTANCE**		**RMI** **M (IQR)**	**US** **M (IQR)**
CONTROLS	Medial Distance	8.0–10 mm (IQR) (M9)	8.0–10 mm (IQR) (M10)
CASES	Lateral Distance	13.5–18 (IQR) (M16) Mild: <13.5 mm Moderate: 13.5–18 mm Severe: >18 mm	14–18 (IQR) (M16) Mild: <14 mm Moderate: 14–18 mm Severe: >18 mm

**Table 2 diagnostics-13-01496-t002:** (**A**,**B**) Student’s *t*-test. Retinacle Thickness. (**A**). Student’s *t*-test to differentiate the media values of retinacle thickness IN CASES AND CONTROLS using MRI and US. The difference in media values of medial retinaculum thickness in cases and controls is statistically significant using the Student’s *t*-test in both MRI and US. MRI (Media cases = 1.98 ± 0.67 mm; e media controls = 0.99 ± 0.17 mm). US (Media cases = 2.24 ± 0.81 mm; e media controls = 1.05 ± 0.48 mm). The difference in media values of lateral retinaculum thickness in cases and controls is statistically significant using Student’s *t*-test in both MRI and ECO. MRI (Media cases = 1.8 ± 0.6 mm; e media controls = 0.86 ± 0.14 mm). US (Media cases = 2.01 ± 0.67 mm; e media controls = 1.03 ± 0.58 mm). (**B**). Student’s *t*-test to compare the media values of MRI AND US to evaluate retinacle thickness in cases and controls. Media values of retinacle thickness examined by US and MRI show statistically significant differences (Retinacles: Media MRI: 1.98 ± 0.67 mm, Media US: 2.24 ± 0.10 mm). In controls, there was no statistically significant difference.

(**A**)			
**RENITACLES**		**Mean Values**	**St Deviation**
Medial Retinaculum *p* < 0.001	CONTROLS	MRI 0.995 US 1.05	MRI 0.1708951 US 0.4893153
	CASES	MRI 1.984 US 2.241667	MRI 0.671581 US 0.811212
Lateral Retinaculum *p* < 0.001	CONTROLS	MRI 0.8675 US 1.03	MRI 0.143915 US 0.5823857
	CASES	MRI 1.855 US 2.016167	MRI 0.6176568 US 0.6705325
(**B**)			
**THICKNESS MEDIAL RETINACULUM**		**Mean Value**	**St Deviation**
CASES *p* < 0.001	MRI	1.984	0.671581
	US	2.241667	0.811212
CONTROLS *p* = 0.4205	MRI	0.995	0.1708951
	US	1.0575	0.4893153
**THICKNESS LATERAL RETINACULUM**		**Mean Value**	**St Deviation**
CASES *p* < 0.001	MRI	1.855	0.617
	US	2.016167	0.670
CONTROLS *p* = 0.0735	MRI	0.86	0.82
	US	1.03	0.84

**Table 3 diagnostics-13-01496-t003:** Student’s *t*-test. Thickness of Throclear Cartilages. Student’s *t*-test used to compare the media values of the thickness of trochlear cartilages examined by US and MRI in cases and controls. The difference in media values of medial and lateral trochlear cartilages in cases and controls is statistically significant using Student’s *t*-test. applied in both MRI and ECO. MRI (Media cases = 1.79 ± 0.50 mm; media controls = 3.83 ± 0.88 mm). US (Media cases = 1.62 ± 0.52 mm; media controls = 3.56 ± 1.0 mm). MRI (Media cases = 2.2 ± 0.47 mm; media controls = 4.3 ± 0.83 mm). US (Media cases = 2.0 ± 0.47 mm; media controls= 3.9 ± 1.0 mm).

THICKNESS CARTILAGES		Mean Value	St Deviation
Medial Cartilage *p* < 0.001	CONTROLS	MRI 3.8375 US 3.5625	MRI 0.883956 US 0.159031
	CASES	MRI 1.793833 US 1.626667	MRI 0.5065774 US 1.491565
Lateral Cartilage *p* < 0.001	CONTROLS	MRI 4.3075 US 3.9575	MRI 0.8398374 US 1.059726
	CASES	MRI 2.225 US 2.0575	MRI 1.4735531 US 1.4787718

**Table 4 diagnostics-13-01496-t004:** (**A**,**B**) Student’s *t*-test. Patello-Femoral Distance. (**A**). Student’s *t*-test to differentiate medial patello-femoral distance in cases and controls in both MRI and US. The difference in media values of medial patello-femoral distance in cases and controls is statistically significant using the Student’s *t*-test. Medial Patello-Femoral Distance. MRI (Media cases = 15.9 ± 2.72 mm e media controls= 8.9 ± 1.59 mm). US (Media cases= 16.2 ± 2.72 mm e media controls = 7.0 ± 1.36 mm). (**B**). Student’s *t*-test to compare the media values of MRI AND US to evaluate patello-femoral distance in cases and controls. Overlapping of results obtained in cases using MRI and US. No statistically significant differences. Medial patello-femoral distance (Media MRI: 15.9 ± 2.72 mm, Media US: 16.2 ± 2.72 mm). Controls show a statistically significant difference. Medial patello-femoral distance (Media MRI: 8.89 ± 1.61 mm, Media US: 9.2 ± 1.36 mm).

(**A**)			
**PATELLO-FEMORAL DISTANCE**		**Mean Value**	**St Deviation**
MEDIAL DISTANCE	CONTROLS	MRI 8.9 US 9.205128	MRI 1.598076 US 1.360717
	CASES	MRI 15.91667 US 16.23333	MRI 2.726466 US 2.726932
(**B**)			
**MEDIAL PATELLO-FEMORAL DISTANCE**		**Mean Value**	**St Standard**
CONTROLS *p* < 0.0001	MRI	8.89	1.61
	US	9.20	1.36
CASES *p* < 0.0658	MRI	15.91	2.72
	US	16.2	2.72

**Table 5 diagnostics-13-01496-t005:** Descriptive analysis of all clinical data. Median and Interquartile Range values of all clinical data analyzed (KOOS, function daily living, function sports and recreational activity, quality of life) in healthy controls and pathological cases.

**KOOS**	
Controls	0.8–0.87 (IQR) (M0.84)
Cases	0.37–0.68 (IQR) (M0.52)
**SYMPTOMS STIFFNESS**	
Controls	0.89–0.93 (IQR) (M0.93)
Cases	0.37–0.70 (IQR) (M0.58)
**FUNCTION, DAILY LIVING**	
Controls	0.79–0.85 (IQR) (M0.82)
Cases	0.37–0.66 (IQR) (M0.54)
**FUNCTION SPORTS AND RECREATIONAL ACTIVITY**	
Controls	0.75–0.75 (IQR)(M0.75)
Cases	0.38–0.60 (IQR) (M0.50)
**QUALITY OF LIFE**	
Controls	0.75–0.88 (IQR) (M0.88)
Cases	0.16–0.54 (IQR) (M0.45)

**Table 6 diagnostics-13-01496-t006:** Student’s *t*-test to compare media values of all clinical data in cases and controls. All media values are statistically significant.

	Mean Value	St Deviation
**KOOS**		
Controls	0.84475	0.0725714
Cases	0.5095	0.165666
**SYMPTOMS STIFFNESS**		
Controls	0.89675	0.0663861
Cases	0.5433333	0.1829946
**FUNCTION, DAILY LIVING**		
Controls	0.8085	0865448
Cases	0.528	1471515
**FUNCTION SPORTS AND RECREATIONAL ACTIVITY**		
Controls	0.7823077	0.0703548
Cases	0.4583333	0.180011
**QUALITY OF LIFE**		
Controls	0.8148718	0.0879555
Cases	0.4003333	0.2148627
**VAS**		
Controls	0.94825	0.90437
Cases	6.366667	1.850

**Table 7 diagnostics-13-01496-t007:** Correlation Matrix to compare instrumental and clinical data. Correlation results of clinical and instrumental parameters showing high direct or inverse correlation. A correlation between medial condyle distance measured with MRI and US and the VAS scale is direct and equal to 99% (0.9873), statistically significant (*p* < 0.001). The correlation between medial condyle distance measured with MRI and US and the KOOS score is inverse and equal to 98% (−0.9814), statistically significant (*p* < 0.001).

MEDIAL PATELLO-FEMORAL DISTANCE		
*p* < 0.001	**US**	**MRI**
**VAS**	0.9873	0.9746
**SYMPTOMS STIFFNESS**	−0.9552	−0.9649
**KOOS**	−0.9814	−0.9649
**FUNCTION, DAILY LIVING**	−0.9252	−0.9042
**QUALITY OF LIFE**	−0.9471	−0.9191
**FUNCTION SPORTS AND RECREATIONAL ACTIVITY**	−0.8266	−0.8340

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
