# Peer review of "Patello-Femoral Pain Syndrome: Magnetic Resonance Imaging versus Ultrasound"

_diagnostics, 2023, doi:10.3390/diagnostics13081496_

Round 1

Reviewer 1 Report

• In the Introduction section, the long version of “SFR” should be included in the text.

• Types and sizes of texts in Tables and Figures must be the same. It should be checked throughout the Manuscript.

• Numerical values in figures should be made legible, and shape quality should be improved.

• The results obtained with the studies belonging to the last years should be compared. There are no references for 2023, 2022, and 2021 in the references.

• The sentences in the study should be original and should not contain the sentences in other works: https://doi.org/10.1179/1753614615Z.00000000099; http://dx.doi.org/10.1136/bjsm.2006.034215; https://doi.org/10.1259/bjr.20170456; https://doi.org/10.1179/1753614615Z.000000000100

Author Response

Dear colleague,

thank you for the valuable suggestions to improve our work, we have made the following changes to the text:

1)in the introduction we replaced SFR with PFS to make the text as clear as possible, always using the same abbreviation;

2-3) we revised tables and figures, standardizing tables and making the figures more legible;

4) we have added more recent bibliographical references;

5) we have significantly modified the introduction and discussion.

Reviewer 2 Report

Manuscript ID: diagnostics-2311387

Patello-Femoral Pain Syndrome: Magnetic Resonance Imaging versus Ultrasound

Thank you for the opportunity to review the manuscript. The review process aims to assess the quality and ensure the article’s reliability, completeness, and consistency. It is a way to improve your manuscript, and I hope you find my comments helpful.

Overall: It is an interesting manuscript. Methodologically, I have nothing important to criticise. The study is interesting, adds novelty to the knowledge of PFS diagnosis, and, with significant changes in its drafting, could be assessed for publication.

Please see specific comments below.

Title: adequate and correct.

Abstract:

Correct in length. A sentence with a clear definition of the study’s objective is missing. The authors explain what was done but not explicitly the objective. Nor do the authors clearly state the objective at the end of the introduction.

Introduction:

Excessively long and with information that is optional to focus the issue. I would delete the paragraph from lines 67 to 77. The diagnosis is predominantly clinical, and to date, we have no evidence that any imaging studies provide relevant information for the diagnosis of FPS (this is the concept that you must emphasise). Differential diagnosis is not relevant, and surgical treatment options are not relevant.

The paragraph “Therefore, the aim of the present study is to demonstrate how MRI and US investigation, mainly in combination, could contribute to the diagnosis of patellofemoral instability taking precise measurements of some knee structures as compared with clinical data” is vague and should make the study more concrete. One idea is to answer questions: Is the US better than MRI for diagnosis? Do they measure the same? Are there differences in measurements between cases and controls? Are US and MRI recommended to complete the clinical information? Is the US study sufficient, and can it replace MRI?

Line 124: The Movie Theatre Sign is instead a pain on the front of the knee after sitting for an extended period with the knees bent, such as one does in a movie theatre or when riding on an aeroplane. Pain rising after prolonged sitting is another clinical feature.

I want to remind you of some tips on the introduction: It introduces the topic but does not develop it. Its purpose is to identify the question being investigated: the WHAT and WHY of the study. It defines the problem and relates the background according to the evidence provided by previous studies. It should explain to the reader the current state of knowledge and how the study is justified. In summary, it contextualises and documents the need for the manuscript. As a final paragraph, the introduction should include a clear and precise description of the study’s hypothesis (if applicable) of the hypothesis of the study (if there was one) and the objectives of the work (at least the primary or at least the primary or main objective).

Materials and methods:

The description is correct. It is long, but paragraphs can only be shortened, losing information about the study’s development.

Results:

A correct presentation of results in the tables is done. This is the best way to see the results.

Discussion:

The two opening paragraphs (lines 349-366) can be omitted. They are more suited to being in an introduction. The discussion usually starts with briefly describing the most relevant results and their significance. It should interpret the results of the study.

You should rewrite the discussion (just as you should rewrite the introduction better). The manuscript can be significantly improved. The research is interesting but needs to be better reported so the reader can appreciate it.

References: they are correct.

Figures: correct and adequate.

Author Response

Dear colleague,

thank you for the valuable suggestions to improve our work, we have made the following changes to the text:

ABSTRACT:

we have included this sentence in the abstract: "purpose of the study: to evaluate the diagnostic role of MRI and US findings associated with PFS defining the range values of instrumental measurements obtained in pathological cases and healthy controls, the performance of the two methods in comparison and the correlation with clinical data."

INTRODUCTION AND DISCUSSION:

we have rewritten the following sections as requested.

Round 2

Reviewer 2 Report

You have modified what was requested of you. The manuscript is interesting, and it is publishable.